# A comparative study of thermal radiation model in Chinese solar greenhouse

**Wenhe Liu**[1⊚], **Manhe Qu**[1⊚], **Feng Zhang**[1]*, **Simin Cao**[2], **Zehui Li**[1], **Zhanyang Xu**[1]

**1** University of Shenyang Agricultural, Shenyang city, Liaoning Province, China, **2** China Nuclear Power Research and Design Institute, Chengdu, China

⊚ These authors contributed equally to this work.
* Fzhang2019@syau.edu.cn

**Data Availability Statement:** All relevant data are within the manuscript and its Supporting Information files.

## Abstract

The performance of three widely used thermal radiation models, the P-1 model, the surface-to-surface (S2S) model and the Discrete-Ordinates (DO) model, were evaluated for simulation temperature in Chinses solar greenhouse. The thermal radiation models were evaluated by comparing the numerical results with experimental data at representative points in the CSG. For indoor rear wall, the indoor soil and indoor air, the models showed good agreement between the experimental data and the simulated results correspond to P1, S2S and DO respectively. This work provides information for simulate greenhouse temperature and use specific radiation models for the most suitable thermal environment for crop growth.

## Introduction

Chinese solar greenhouse(CSG) is mainly developed to create a suitable indoor environment to meet the temperature, humidity, lighting and other needs of crop's growth by uses the energy of the sun [1, 2]. The operation principle of a solar greenhouse is mainly based on the utilization of solar energy. It allows sunlight to enter the interior of the greenhouse through transparent materials such as glass or plastic. These materials not only allow light to pass through, but also absorb incoming light energy and convert it into thermal energy, thereby increasing the temperature inside the greenhouse. During the day, sunlight shines into the greenhouse, and transparent materials accumulate heat. At night, when the outdoor temperature drops, the insulation of the greenhouse will be turned off to reduce the loss of indoor heat. Meanwhile, relying on the slow release of heat from walls and soil at night, indoor temperatures can be maintained at a relatively high level, thereby supporting plant growth. The problem of vegetable off-season supply is solved by historic contributions of CSG in the northern China's winter, which is benefit for increasing farmers' income, saving energy, avoiding environmental pollution caused by greenhouse heating and stabilizing society [3]. Therefore, CSG has been widely used. China is located in the East Asian monsoon climate zone, the Chinese climate is cold in winter and hot in summer, which is necessary to regulate and control the increasing and decreasing temperature in winter and summer respectively [4, 5]. The information of climate and environment inside greenhouse is also increasingly important for greenhouse environment control. Therefore, the study of solar greenhouse thermal radiation

**Funding:** This study was supported by the Postdoctoral Research Foundation of China (2021M693862) awarded to Feng Zhang."

**Competing interests:** The authors have declared that no competing interests exist.

provides a theoretical basis for greenhouse environmental control, which is a good research value and practical application prospect.

The computational fluid dynamics (CFD) method can be used to simulate the climate environment of indoor CSG. Coussirat et al. [6] studied the performance of several turbulence and radiation models to simulate the behavior of double glazed façades, and compared the obtained results with experimental data taken from the other literature. The RNG k–ε turbulence model is better than the other turbulence models tested in terms of predicting heat transfer, zones of low velocities within the façade configuration. Dynamic simulations of climatic parameters inside the planting greenhouse were carried out. Radiative heat transfers were modeled using a bi-band DO model, and the crop was considered to be a porous medium [7]. The results of the heterogeneity of temperature and transpiration flux within the crop rows and their daily evolution was emphasized. The performance of three widely used turbulence models, the standard k–ε model (SKE), the renormalization group k–ε model (RNG), the realizable k–ε model (RKE), were compared for their ability to predict the airflow velocities and ammonia concentrations in the scale model swine building enclosure. The RNG model was suitable for predicting weak airflows. The velocities and concentrations predicted by the RNG model were closer to the measured values than other two models [8]. In order to study the importance of the greenhouse effect in driving flow and heat transfer characteristics through the buoyancy of the system, a three-dimensional unsteady model with the RNG k-ε turbulence and DO radiation was developed, by using CFD method. The analysis showed that simulating the greenhouse effect had an important role to accurately predict the characteristics of the flow and heat transfer in solar chimney power plant systems [9]. The radiative exchanges were taken into account by using the Solar Load Model available in ANSYS Fluent, which was the combination of a solar ray tracing algorithm and a radiation model called surface-to-surface (S2S) [10]. The simulation results showed that predicting the microclimate can contribute to enhanced performance in these kinds of greenhouses by improving the radiation transmission efficiency inside. Zhang Fang et al. [11] established a large-span greenhouse based on CFD technology and used DO radiation model to simulate the temperature and airflow field under natural ventilation conditions. In addition, the absolute error between the model simulation value and the measured value is $0.2 \sim 2.8\degree C$. The root mean square error is $1.6\degree C$. The maximum relative error is 9.9%. The average relative error is 4.1%. Therefore the numerical simulation results were in good agreement with the measured values by experimental verification. Xiangdong Li et al. [12] studied S2S models to simulated thermal radiation in indoor spaces. The radiative effects were introduced in the thermal flowing models as a critical condition for indoor temperature environments. The P1 radiation model was used for exploration of the structure parameters of CSG by CFD simulation technology [13]. The simulation results of the model were basically corresponding to the actual measurement. In order to improvement of air flow distribution, air velocity and temperature inside a mixed greenhouse dryer was numerically investigated by using 3D CFD ANSYS fluent code. The DO model was used to simulate the mechanism of heat transfer from solar radiation to the greenhouse. The results showed that installing an air recirculation system, into the greenhouse could increase the air velocity in the drying chamber from 0.71 m/s to 1.5 m/s and the temperature from 315 K to 360 K [14]. Based on CFD technology, the indoor environment of CSG was conducted by using standard k-ε turbulence model and DO radiation model under natural ventilation conditions and natural ventilation conditions respectively [15]. The error between simulation and experiment results were located in a reasonable range. Therefore, the numerical simulation results could be used as a theoretical basis for studying the indoor environment distribution under different ventilation modes. Yuehong Ma et al. [16] simulated the thermal insulation effect of composite wall and brick wall in CSG, which was based on FLUENT software, by

using DO radiation model and standard k-ε turbulence model. The simulation results showed that the thermal performance of the block composite wall CSG was obviously better than that of the brick wall CSG, which was corresponding to the actual cultivation test results under the same condition. During the research of designing parameters' effect on the thermal environment in indoor CSG, the use of CFD codes has been extended to gain insight into this problem, but selecting appropriate sub-models for the influence of convection, radiation and turbulence were still a huge challenge [17]. Based on the results of all previous works, it should be noted that the specific radiation model used to provide the most suitable thermal environment for crop growth has not been researched. Therefore, it is necessary to discussion the application of radiation model in CSG simulation.

Due to cost limitations of greenhouses, most greenhouses in Northeast China are single slope sunlight greenhouses, and the temperature inside the sunlight greenhouses is an important indicator affecting crop growth. Due to the influence of solar radiation, the temperature inside the greenhouse is too high at noon, often exceeding the suitable growth temperature for crops. This study conducted CFD numerical simulations on solar greenhouses in Northeast China and analyzed the effects of three radiation models (P1, S2S, and DO) on predicting the temperature environment inside the greenhouse. The purpose is to study the ability of different radiation models to predict the temperature environment inside greenhouses, in order to make more accurate numerical predictions of the climate conditions inside solar greenhouses.

## Experimental method and theoretical considerations

### Experimental model

To validate the CFD model, real-scale experimental data were used. The solar greenhouse used is located in Northeast China (latitude:41°49′N, longtitude:123°34′). CSG is composed of north wall, east wall, west wall, plastic film and ground soil without heating measurement in winter. The greenhouse is a length of 60m, a span of 8m, a rear wall height of 3.2m and a ridge height of 5m respectively. The rear wall is composed of 240mm clay bricks, 120mm polyethylene benzene board, and 240mm clay bricks. The rear slope is composed of 15mm wooden boards, 150mm polyethylene benzene board, 25mm cement, and 90mm waterproof layer. The front roof is made of 0.1mm PVC anti-aging plastic film for daytime lighting and heat storage. At night, the insulation covering materials are 1.5mm PE woven fabric, 27mm spray glue cotton, and 1.5mm PE woven fabric from the inside out. The spray glue cotton plays a role in insulation, while the inner and outer layers of PE woven fabric play a waterproof role. It is opened at 8:00 and covered at 16:00 in every day. Fig 1 shows the indoor and outdoor conditions of the experimental greenhouse. During the experiment, strawberries were planted in the greenhouse, with an average plant height of 20cm, without obstructing the back wall of the greenhouse.

### Testing arrangement

During the experiment, the TRM-ZS3 greenhouse environment acquisition system of Jinzhou Sunshine Technology Co., Ltd. is used to monitor the internal temperature of CSG(Temperature measurement range: -40°C∼80°C, measurement progress range: ± 0.4°C), the ADCON instrument is used to monitor the soil temperature, which is made in the United States(Temperature measurement range: -40°C∼120°C, measurement progress range: ± 0.2°C). The TRM-ZSFGPRS wireless remote control system from Jinzhou Sunshine Technology Co., Ltd. is used to monitor outdoor temperature(Temperature measurement range: -25°C∼70°C). The specific instruments are shown in Fig 2.

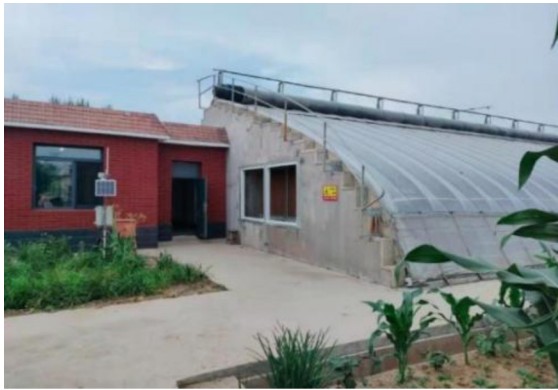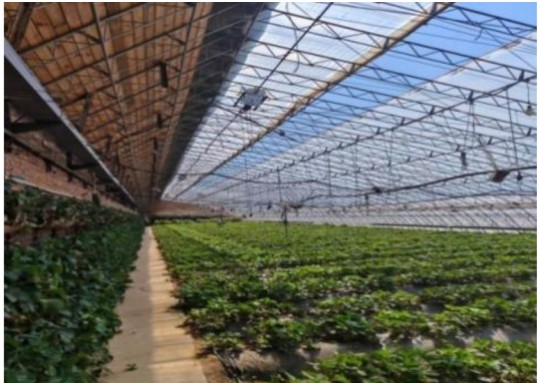

**Fig 1. Experimental solar greenhouse.**

Indoor temperature data of the CSG is recorded by test instruments every 10min, while outdoor temperature data of the CSG is recorded in every 15min of all days. A total of 8 probes are arranged. The location of each measuring point is shown in Fig 3. The temperature sensors are arranged using the cross-sectional method. In the greenhouse, four temperature measurement points are arranged in two layers on a section 5m away from the south wall. Three temperature measurement points (T1, T4, T6) are arranged at a height of 1m from the ground on the bottom layer, and one temperature measurement point (T3) is arranged at a height of 2m from the ground on the upper layer. Three temperature measurement points (T2, T5, T7) are arranged on the soil inside the greenhouse. One rear wall temperature measurement point (T8) is arranged at a height of 1.6m from the ground, and the temperature measurement points are basically not affected by external wind. The current research season is winter. Many factors of condition are considered, such as outdoor air, indoor air, wall material parameter, maintenance structure material parameter, soil parameter, etc.

## Geometrical model, materials and mesh

The computational fluid dynamics geometric model of the temperature environment in the solar greenhouse is established, as shown in Fig 4. The positive direction of Z axis is east. The positive direction of X axis is north. The size of the model is the same as that of the real CSG, and the soil thickness of the model is 1m.

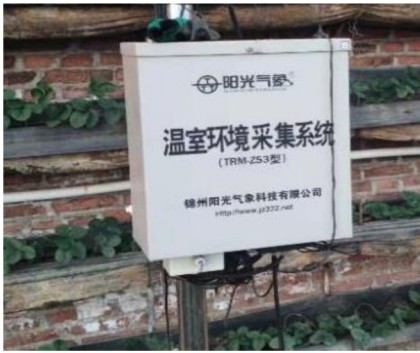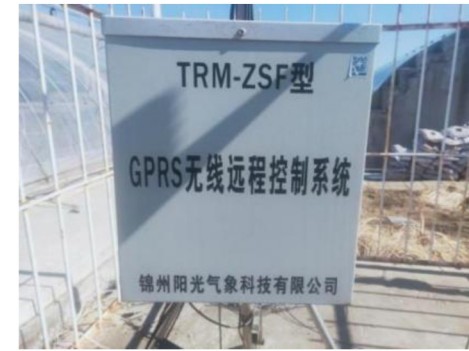

Greenhouse ambient temperature collector                    Outdoor weather station

**Fig 2. Test instrument.**

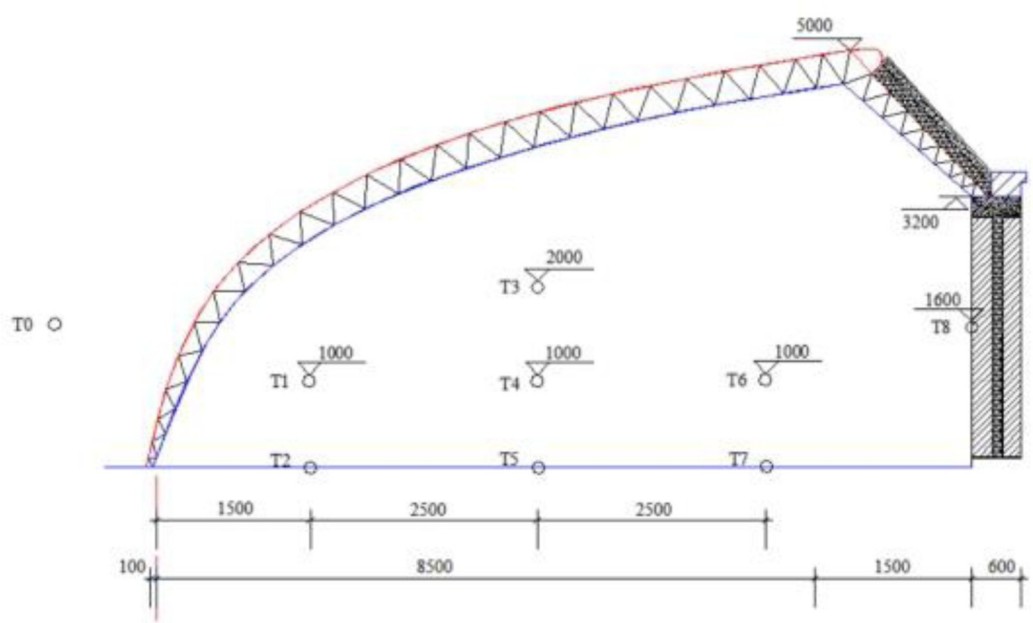

**Fig 3. Test equipment arrangement.**

The grid generation is based on the mesh module built in ANSYS-workbench. The sweeping method is used to divide the hexahedral grid. The basic size of the grid is 0.3m. A total of 142509 nodes and 129600 grid cells are generated. The grid generation of the CSG is shown in

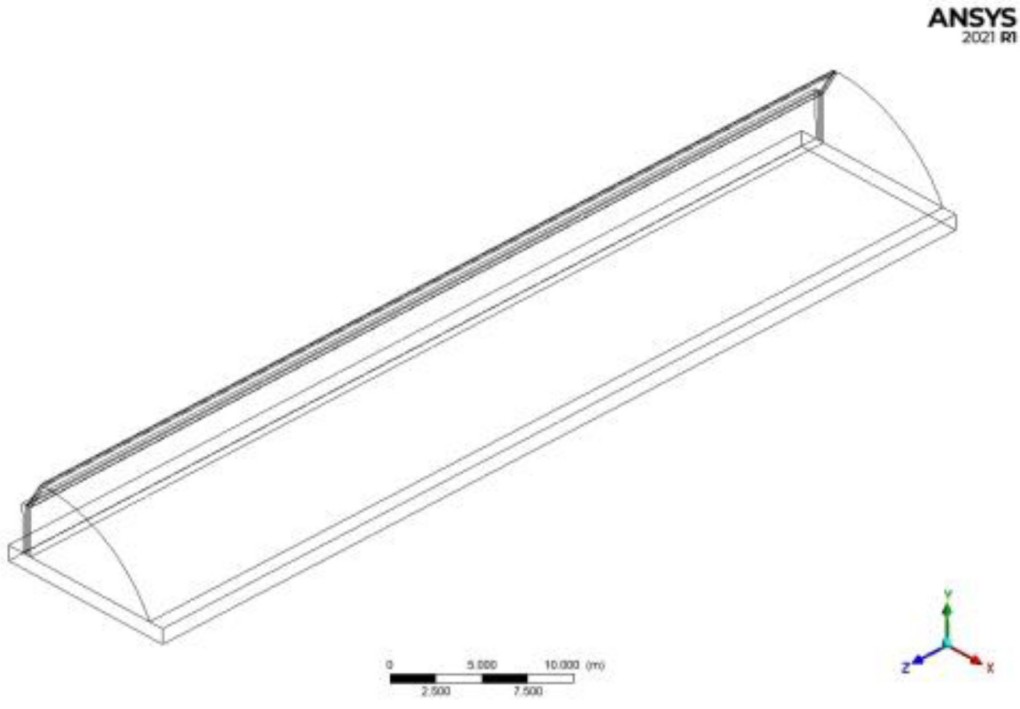

**Fig 4. Geometric model of solar greenhouse.**

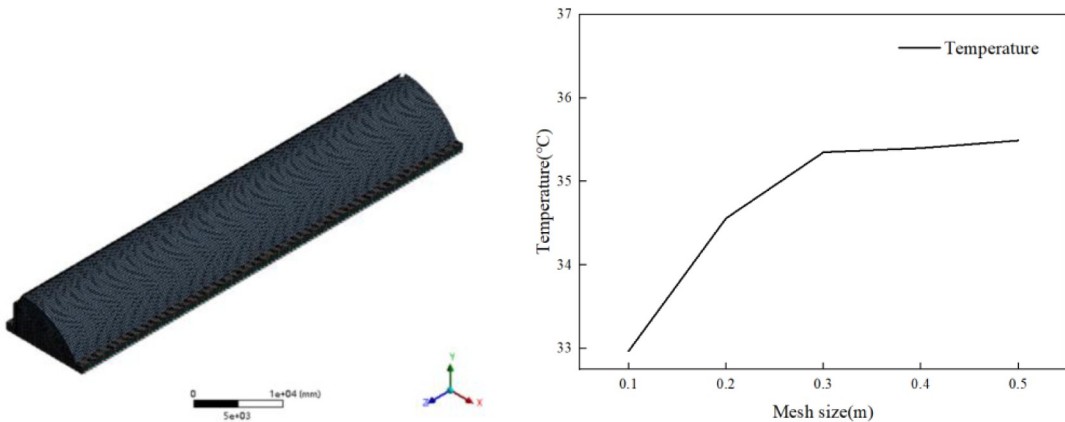

**Fig 5. Grid division of solar greenhouse and the mesh convergence study.**

Fig 5. The grid division result is checked by the distortion inspection standard. The grid distortion is below 0.6. The grid division standard is satisfied according to the grid distortion standard. Under the same numerical simulation conditions, simulation calculations were conducted and point T4 was selected as the temperature verification feature point. The consistency and stability of the simulation results were compared. The relationship between the simulated temperature of the feature point and the grid size is shown in Fig 5. When the grid size is less than 0.3m, the temperature of the feature point fluctuates significantly. When the grid size is greater than 0.3m, the temperature of the feature point does not change significantly with the increase of the grid size and remains relatively stable. The values of thermophysical parameters of building materials are shown in Table 1.

## Boundary condition

Import the grid file into the computational fluid dynamics software ANSYS Fluent 2022R2 for numerical solution. Fluent provides rich computational models and uses the finite volume method to solve the computational domain. In this study, the greenhouse film is treated as semi-transparent boundaries, and the soil is treated as opaque boundaries. The absorption and transmittance of solar radiation on each surface are set, and specific optical parameters are shown in Table 2. The radiation boundary conditions for greenhouse plastic films need to consider the impact of environmental radiation. Based on experimental data, the benchmark temperature for the early morning environment is set at 11°C, and the solar radiation factor for March was determined based on historical meteorological data and relevant research in Shenyang, to match the simulated and measured temperature time series. In addition, boundary conditions for greenhouse simulation are set in combination with actual measurement conditions. The external temperature values of the greenhouse at different times are used as the environmental convection boundaries for the enclosure structure and outdoor soil. The temperature values are shown in Fig 6. Using temperature values at different times as the

**Table 1. Properties of building materials [18–20].**

| Attributes (units) | Common brick | Cement | Soil | Plastic film | Polystyrene foam board | Quilt |
|---|---|---|---|---|---|---|
| Density(kg/m3) | 1700 | 2300 | 880 | 920 | 50 | 50 |
| Specific heat capacity(j/(kg·K)) | 850 | 880 | 1170 | 2100 | 880 | 1000 |
| Coefficient of thermal conductivity(w/(m·k)) | 0.42 | 1.50 | 0.94 | 0.047 | 0.027 | 0.05 |

**Table 2. Optical parameters of materials in greenhouses.**

| materials/position | Boundary type | Absorption coefficient($m^{-1}$) | emissivity | absorptivity | transmissivity |
|---|---|---|---|---|---|
| air | transparent | 0.50 | - | - | - |
| Greenhouse film | semi-transparent | - | 0.85 | 0.95 | 0.95 |
| Soil surface | opaque | - | 0.95 | 0.6 | - |

environmental convection boundary of the greenhouse film, the temperature values are shown in Fig 7. In order to solve the heat transfer process of solar greenhouse under 24h sunlight, the solution step is set as 1800s, and the total solution time is 86400s.

## Governing equation

Simulating the distribution of temperature and airflow field, etc. in a greenhouse, the compressibility of a gas moving at low speeds can usually be solved as an incompressible fluid when it does not have much effect on its motion and equilibrium problems. The air flow rate in the solar greenhouse is a low-flow rate flow, and the size of the flow rate is usually around $0.1 \sim 0.5 m/s$. Therefore, the basic control equations satisfied by the fluids in the greenhouse include the mass conservation equation, the momentum conservation equation, the energy conservation equation, and the generalized form of the control equations.

(1) Continuity equation:

$$\frac{\partial \rho}{\partial t} + \frac{\partial (\rho u_i)}{\partial x_i} = S_m \tag{1}$$

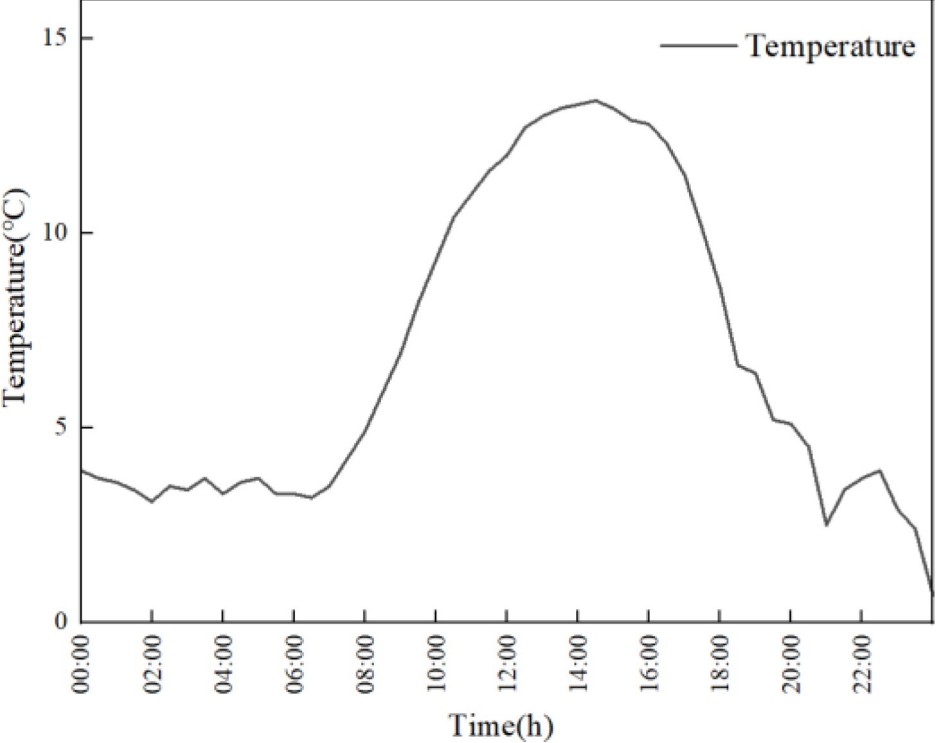

**Fig 6. Ambient temperature outside the greenhouse.**

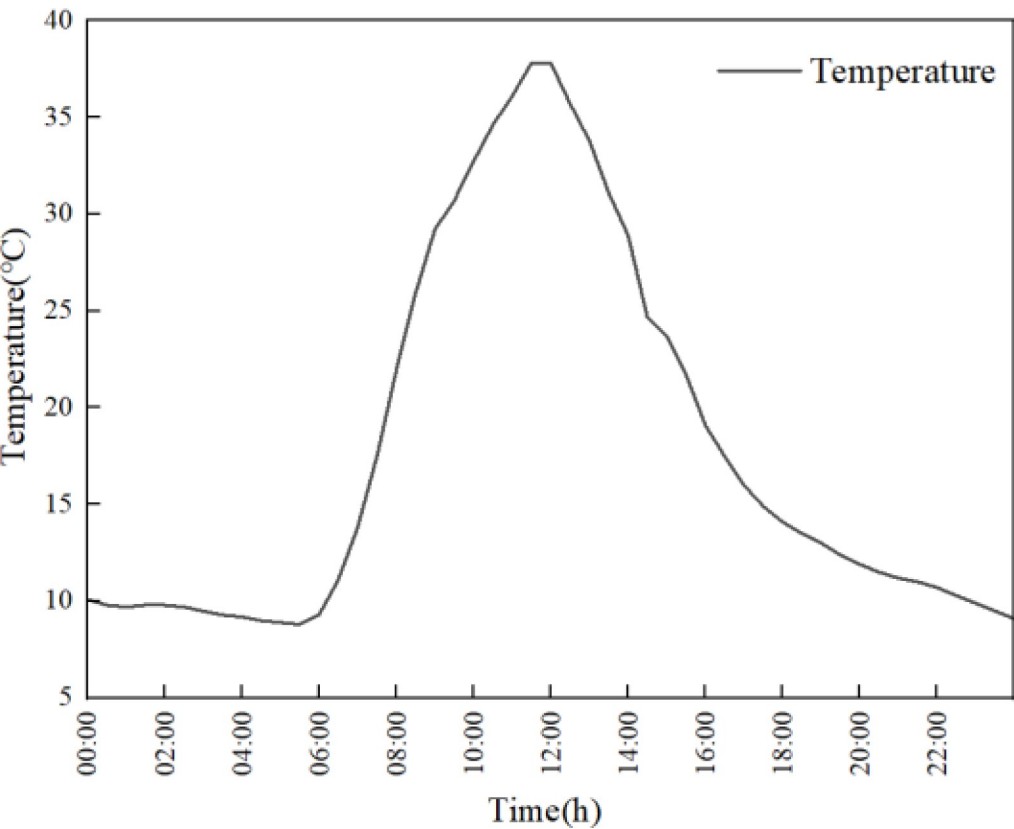

**Fig 7. Greenhouse plastic film temperature.**

where $\rho$ is the density of the gas. $u_i$ represents the component of the velocity vector in the x,y,z direction. $S_m$ is the mass of the secondary phase added to the continuum term, which can be any custom source term.

(2) Conservation of momentum equation:

$$\frac{\partial(\rho u)}{\partial t} + div(\rho u U) = -\frac{\partial P}{\partial x} + \frac{\partial \tau_{xx}}{\partial x} + \frac{\partial \tau_{yx}}{\partial y} + \frac{\partial \tau_{zx}}{\partial z} + F_x \tag{2}$$

$$\frac{\partial(\rho v)}{\partial t} + div(\rho u U) = -\frac{\partial P}{\partial x} + \frac{\partial \tau_{xy}}{\partial x} + \frac{\partial \tau_{yy}}{\partial y} + \frac{\partial \tau_{zy}}{\partial z} + F_y \tag{3}$$

$$\frac{\partial(\rho w)}{\partial t} + div(\rho u U) = -\frac{\partial P}{\partial x} + \frac{\partial \tau_{xz}}{\partial x} + \frac{\partial \tau_{yz}}{\partial y} + \frac{\partial \tau_{zz}}{\partial z} + F_z \tag{4}$$

where P is the pressure on the fluid micromeres in Pa. $\tau$ is the viscous stress on the micromeres. F is the different micromeres viscous components in the corresponding.

(3) The conservation of energy equation:

$$\frac{\partial(\rho E)}{\partial t} + \frac{\partial}{\partial x_i}(u_i(\rho E + p)) = \frac{\partial}{\partial x_i}\left(k_{eff}\frac{\partial T}{\partial x_i} - \sum_{j'} h_{j'} J_{j'} + u_j\left(\tau_{ij}\right)_{eff}\right) + S_h \tag{5}$$

where $k_{eff}$ is the effective heat transfer coefficient, $k_{eff} = k + k_t \cdot k_t$ is the turbulent heat transfer

coefficient, $E = h - \frac{p}{\rho} + \frac{u_i^2}{2}$ is the total energy. $\frac{\partial}{\partial x_i} \sum_{j'} h_{j'} J_{j'}$ is the change in enthalpy of the humid air transport process. $S_h$ is the volumetric heat source term.

## Turbulence models

The mass, momentum and energy conservation equations are solved by using a finite volume method in CFD calculation. The details of the equations and solution procedures can be found in the software documents [21]. However, no single model is universally accepted as being superior for all classes of problems. The choice of the turbulence model depends on multiple considerations, including the physics of the flow, the level of accuracy required, the available computational resources and the amount of time available for the simulation [22]. The transport equations and model constants for the turbulence models are listed in Table 3 [8].

The RNG K-ε model is advantaged by high stability, economy, small calculation workload and high accuracy, which is provided in FLUENT [23, 24]. It can reasonably predict large-scale turbulence and has the widest application range in industrial flow and heat transfer simulation [25, 26]. In current study, the gas in the middle temperature chamber is set as an incompressible ideal gas, so the RNG K-ε model is used to simulate the flow state of the flow field in the greenhouse.

## Radiation models

To model the thermal radiation CSG, it is essential to determine a reliable radiation model. For radiation simulation, five radiation models, including S2S radiation model (Surface to surface model), Rosseland radiation model, P-1 radiation model, DTRM radiation model (Discrete transfer method) and DO radiation model (Discrete ordinates model) are provided by ANSYS fluent. The solar radiation model is compatible with the five radiation models in Fluent. The parallel solver was used. The DTRM model works for all optical thicknesses, but does not take into account the scattering effect of the radiation in the calculation, which increases the burden on the CPU if ray tracing is to be used. The Rosseland radiation model is applicable to the calculation of models with optical thickness>5, and is generally applicable to the application of glass models. Therefore, this study uses P1, S2S and DO heat transfer models to calculate solar radiation.

The full name of the S2S radiation model is Surface to Surface radiation model, which is a model that only considers thermal radiation between surfaces. It is a radiation model suitable for zero optical thickness scenarios. If the medium in the computational domain is some medium with strong absorption of thermal radiation, such as water vapor (polyatomic

**Table 3. Transport equations and model constants for the turbulence models [8].**

| Model | Transport equations | Model constants |
|---|---|---|
| SKE | $\frac{\partial(\rho k)}{\partial t} + \frac{\partial(\rho k u_i)}{\partial x_i} = \frac{\partial}{\partial x_j}\left(\left(\mu + \frac{\mu_t}{\sigma_k}\right)\frac{\partial k}{\partial x_j}\right) + G_k - \rho\varepsilon$ | $C_{1\varepsilon} = 1.44$ |
| | $\frac{\partial(\rho\varepsilon)}{\partial t} + \frac{\partial(\rho\varepsilon u_i)}{\partial x_i} = \frac{\partial}{\partial x_j}\left(\left(u + \frac{u_t}{\sigma_\varepsilon}\right)\frac{\partial\varepsilon}{\partial x_j}\right) + \frac{c_{1\varepsilon}\varepsilon}{k}G_k - C_{2\varepsilon}\rho\frac{\varepsilon^2}{k}$ | $C_{2\varepsilon} = 1.92$ |
| RNG | $\frac{\partial}{\partial t}(\rho k) + \frac{\partial}{\partial x_i}(\rho k u_i) = \frac{\partial}{\partial x_j}\left(\left(\mu + \frac{\mu_t}{\mu_k}\right)\frac{\partial k}{\partial x_j}\right) + G_k + G_b - \rho\varepsilon - Y_M + S_K$ | $G_{1\varepsilon} = 1.42$ |
| | $\frac{\partial}{\partial t}(\rho\varepsilon) + \frac{\partial}{\partial x_i}(\rho\varepsilon u_i) = \frac{\partial}{\partial x_j}\left(\left(\mu + \frac{\mu_t}{\mu_\varepsilon}\right)\frac{\partial\varepsilon}{\partial x_j}\right) + G_{1\varepsilon}\frac{\varepsilon}{k}(G_k + G_{3\varepsilon}G_b) - G_{2\varepsilon}\frac{\varepsilon^2}{k} - R_\varepsilon + S_\varepsilon$ | $G_{2\varepsilon} = 1.68$ |
| RKE | $\frac{\partial(\rho k)}{\partial t} + \frac{\partial(\rho k u_i)}{\partial x_i} = \frac{\partial}{\partial x_j}\left(\left(\mu + \frac{\mu_t}{\sigma_k}\right)\frac{\partial k}{\partial x_j}\right) + G_k - \rho\varepsilon$ | $C_{1\varepsilon} = 1.44$ |
| | $\frac{\partial(\rho\varepsilon)}{\partial t} + \frac{\partial(\rho\varepsilon u_i)}{\partial x_i} = \frac{\partial}{\partial x_j}\left(\left(\mu + \frac{\mu_t}{\sigma_\varepsilon}\right)\frac{\partial\varepsilon}{\partial x_j}\right) + \frac{c_{1\varepsilon}\varepsilon}{k}G_k - C_{2\varepsilon}\rho\frac{\varepsilon^2}{k}(3.7b)$ | $C_{2\varepsilon} = 1.9$ |

molecules), it is not suitable to use this model. This model is more suitable for air or diatomic molecules, etc. The S2S model was used to solve the problem of radiation exchange between diffuse gray surfaces. The value of energy transfer between two radiation exchange surfaces depends on their area, distance and relative position of the surface. The influence of area size, distance and relative position on radiation exchange is represented by "view factor". The S2S model assumes that the surfaces involved in radiation exchange and the surface reflection are gray and diffuse respectively. The absorptivity of the gray surface is independent of the reflectivity and the wavelength of the light. According to Kirchhoff's law, the emissivity is equal to the absorptivity. For a diffuse reflective surface, the reflectivity is independent of the outgoing (or incident) direction. Therefore, according to the gray model, if a certain amount of radiant energy is incident on the surface, part of the radiant energy is reflected. Meanwhile, part of the radiant energy is absorbed. Then the rest of the radiant energy is transmitted [27, 28].

The energy flux leaving a given surface is consists of directly emitted and reflected energy. The reflected energy flux is depended on the incident energy flux of the surrounding environment, which is expressed in terms of the energy flux leaving all other surfaces. The energy leaving the surface is [29]:

$$q_{out,K} = \varepsilon_k \sigma T_k^4 + \rho_k q_{in,k} \tag{6}$$

where $q_{out,K}$ is the energy flux leaving the surface, W/m². $\varepsilon_k$ is the emissivity. σ is the Stephen Boltzmann constant. $q_{in,k}$ is the radiant energy flux incident to the surface by the surrounding environment, W/m².

The incident energy from another surface to another surface is a direct function of the surface to surface "view factor". The "view factor" is the ratio of the radiant energy incident on K surface to the energy leaving of J surface. The incident energy flux is expressed as the energy flux leaving all other surfaces [30]:

$$A_k q_{in,k} = \sum_{j=1}^{N} A_j q_{out,j} F_{jk} \tag{7}$$

where $A_k$ is the K Surface area. $F_{jk}$ is the view factor between k and j surfaces.

For N surfaces, according to the calculation formula of view factor, the formula is expressed as follow:

$$A_j F_{jk} = A_k F_{kj} \ j = 1, 2, 3, \ldots N \tag{8}$$

Therefore:

$$q_{in,k} = \sum_{j=1}^{N} F_{kj} q_{out,j} \tag{9}$$

$$q_{out,k} = \varepsilon_k \sigma T_k^4 + \rho_k \sum_{j=1}^{N} F_{kj} q_{out,j} \tag{10}$$

It was also written as:

$$J_k = E_k + \rho_k \sum_{j=1}^{N} F_{kj} J_j \tag{11}$$

where $J_k$ is the k energy released or radiated from the surface, W/m²; $E_k$ is the transmission power of k surface, W/m².

The main solution of DO radiation model is proposed to discretize the direction of radiation intensity and solve the radiation equation of each particle in the whole space after discretization. The more discretization times, the more accurate the simulation results will be, but the

computational complexity will also increase. This study used a 3D model to solve a total of 8 quadrants with values theta divisions and phi divisions set to 5. The radiation transport equation is converted into the radiation intensity transport equation in the space coordinate system by the DO radiation model, that is, the radiation equation (RTE) propagating in the $\vec{s}$ direction is regarded as the field equation [31]:

$$\nabla \cdot \left(I\left(\vec{r}, \vec{s}\right)\vec{s}\right) + (a + \sigma_s)I\left(\vec{r}, \vec{s}\right) = an^2\frac{\sigma T^4}{\pi} + \frac{\sigma_s}{4\pi}\int_0^{4\pi} I\left(\vec{r}, \vec{s'}\right)\Phi\left(\vec{r}, \vec{s'}\right)d\Omega' \quad (12)$$

Radiant heat transfer obtained on the wall:

$$q_{in} = \int_{n>0} I_{in}s \cdot n d\Omega \quad (13)$$

Net radiant heat leaving the wall:

$$q_{out} = (1 - \varepsilon_w)q_{in} + n^2\varepsilon_w\sigma T_w^4 \quad (14)$$

where $n$ is the refractive factor of medium in contact with wall. $I$ is the incident radiation. $\Omega$ is the spatial solid angle. $\Phi$ is the phase function, here is 2. $\sigma_s$ is the scattering coefficient. $a$ is the absorption coefficient. $\sigma$ is the constant (Stetan-Boltzman), here is $5.672\times10^{-8}$ W/(m$^2$·K$^4$). $s$ is the stroke length. $\vec{r}$ is the position vector. $\vec{s}$ is the direction vector. $\vec{s}$, is the heat dissipation direction vector.

The discrete coordinates is not in P1 approximation, but the angular space based on spherical harmonic functions is discretized. They are characteristic functions of Laplace operators in spherical coordinates. The P1 approximation method is expressed as the linear term, so it can be concluded that solving the following equation is equivalent to solving the equation [32]:

$$\nabla \cdot (D_{P1}\nabla G) - \kappa(G - 4\pi I_b) = 0 \quad (15)$$

where $D_{p1}$ is the diffusion coefficient of P1, which was defined as:

$$D_{p1} = \frac{1}{3\kappa + \sigma_s(3 - a_1)} \quad (16)$$

Linear Legendre coefficient is used for scattering phase function. Therefore, using P1 approximation, isotropic and linear anisotropic scattering are considered. The second term on the left side of the equation is stand for the radiant heat source $Q_r$. Therefore, only one additional equation is needed to consider radiative transfer.

## Results

The results of the simulations are carried out by monitoring the following: the back wall temperature, the soil temperature and indoor temperature, which is compared with the experimentally measured temperatures. The numerical simulation is conducted by three radiation models, such as P1, S2S and DO model.

### Wall temperature

On March 6, 2023, with clear weather, a comparative analysis was conducted on the temperature data of the greenhouse's rear wall measured by sensors. The predicted rear wall

temperatures of three radiation models are researched under same boundary condition corresponding to the real weather. The temperature-time curve of the relevant data during the day are shown in Fig 8.

Before the insulation is removed from 0:00 to 7:30, the indoor rear wall temperature shows a slow downward trend, reaching the lowest temperature 11.9˚C around 7:30 in the morning. At this time, the numerically simulated result of temperatures by P1, DO, and S2S radiation models are 13.21˚C, 13.32˚C, and 13.05˚C respectively. The temperature error between the experimental and simulation data is 1.31˚C, 1.42˚C and 1.15˚C. After the insulation was removed at 7:30 in the morning, due to the enhancement of outdoor temperature and solar radiation, the indoor temperature rapidly increased and reached its maximum temperature of 37.8˚C at 13:00. The slopes of the P1, DO, and S2S radiation models from the beginning of heating up to the highest temperature are 4.00˚C/h, 3.46˚C/h, and 3.37˚C/h respectively. The P1 model reaching the highest temperature is the fastest, which is followed by DO and S2S. The highest temperature of simulated result in P1, DO and S2S radiation models are 32.93˚C, 32.39˚C and 31.61˚C respectively. The temperature error between experimental and simulation data is 4.87˚C, 5.41˚C and 6.19˚C respectively. From 13:00 to 16:00, due to the rapid decrease in outdoor solar radiation, the temperature of the back wall is also showed a rapid downward trend. When the insulation blanket was recovered at 16:00, the downward trend is performed to slow down.

The error analysis of simulation and experimental results are shown in Fig 9. The maximum errors of simulation results in three different radiation models are 3.24˚C, 3.68˚C and 4.58˚C respectively, and the minimum errors are 0.22˚C, 0.64˚C and 0.77˚C respectively. The temperature profile of the indoor rear wall of the greenhouse fitted by the P1 radiation model best matched the trend of the experimentally measured temperature profile. The maximum error of the simulation results is 24.33%, and the minimum error of the simulation results is 0.93%.For the rear wall of the greenhouse, the temperature predicted by the P1 model is more in line with the measured temperature than other models.

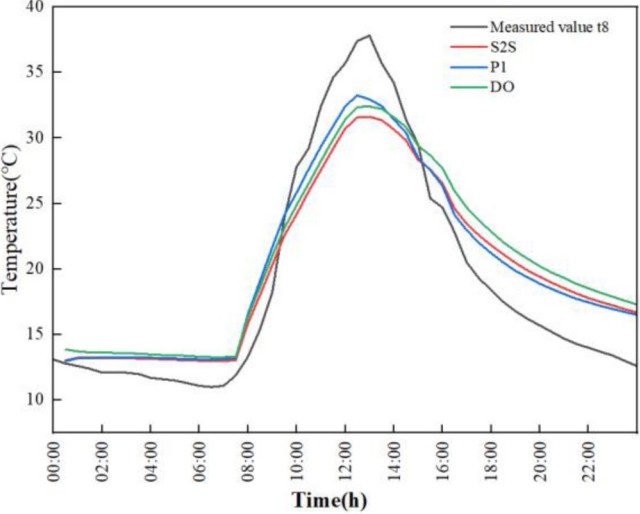

**Fig 8. Experimental and simulated temperature time plots of different radiation models.**

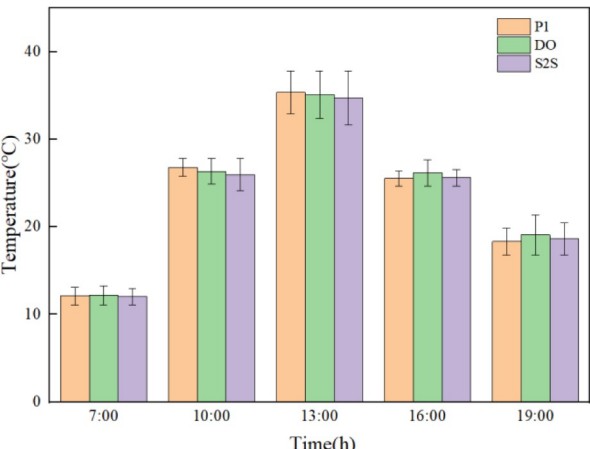

**Fig 9. Comparison error diagram of temperature simulation results and experimental results for the three radiation models.**

## Soil temperature

The evolution of temperature in indoor soil under different positions and radiation models during the whole day are shown in Fig 10.

The overall measured temperature at t2 is reduced due to the distance of the adjacent greenhouse, which is shaded by solar radiation. In this study, point t5 is selected for specific analysis. Before the insulation is removed from 0:00 to 7:30, the experimental temperature of soil shows a slow downward trend. The lowest temperature is 13.3˚C around 7:30 in the morning. The numerically simulated result of temperatures by P1, DO, and S2S radiation models are 13.40˚C, 13.44˚C, and 13.27˚C respectively. The temperature error between the experimental and simulation data is 0.1˚C, 0.14˚C and -0.03˚C. After the insulation is removed at 7:30 in the morning, as a result of the enhancement of outdoor temperature and solar radiation, the indoor temperature is rapidly increased and reached 30.4˚C at 12:30. The slope of the measured experimental data from the beginning of the temperature rise to the highest temperature is 2.88˚C/h. The slopes of the P1, DO, and S2S radiation models from the beginning of heating up to the highest temperature are 3.12˚C/h, 3.19˚C/h and 3.03˚C/h respectively. The DO model reaching the highest temperature is the fastest, which is followed by P1 and S2S. The highest temperature of simulated result in P1, DO and S2S radiation models are 30.57˚C, 30.98˚C and 29.92˚C respectively. The temperature error between experimental and simulation data is 0.17˚C, 0.58˚C and -0.48˚C respectively. From 13:00 to 16:00, the outdoor solar radiation is rapid decreased, the temperature of the soil is also showed a rapid downward trend. When the insulation blanket is recovered at 16:00, the downward trend is performed to slow down.

The error analysis of simulation and experimental results are shown in Figs 11–13 respectively. The maximum and minimum error analysis is shown in Table 4. The soil temperature profile in the greenhouse fitted by the S2S radiation model best matched the trend of the experimentally measured temperature profile. The maximum error of the simulation results is 23.46%, and the minimum error of the simulation results is 0.26%. For the indoor soil of the greenhouse, the temperature predicted by the S2S model is more in line with the measured temperature than other models.

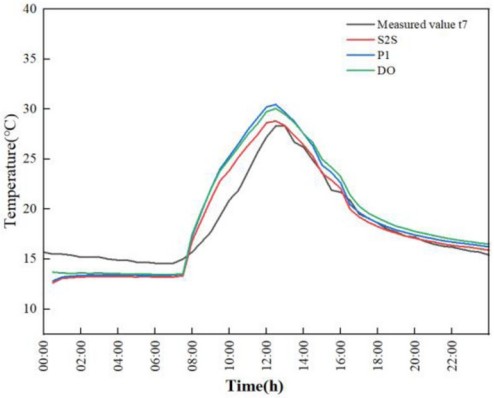

(a) t7 soil point location

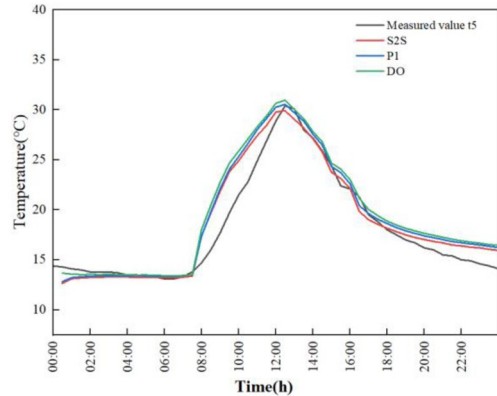

(b) t5 soil point location

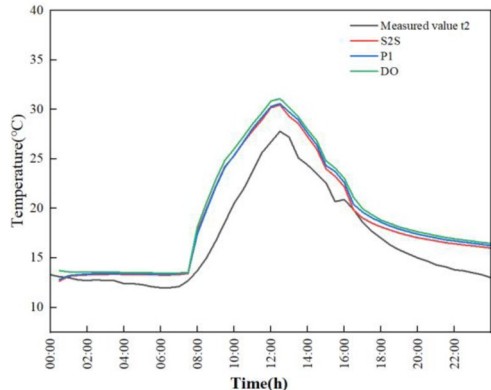

(c) t2 soil point location

**Fig 10. Experimental and simulated temperature time plots for different positions and radiation models.**

## Indoor air temperature

The evolution of temperature in indoor air under different positions and radiation models during the whole day are shown in Fig 14.

In this study, point t6 is selected for specific analysis. Before the insulation is removed from 0:00 to 7:30, the experimental temperature of indoor air shows a slow downward trend. The lowest temperature is 9.9°C around 7:30 in the morning. The simulated temperature has a flat

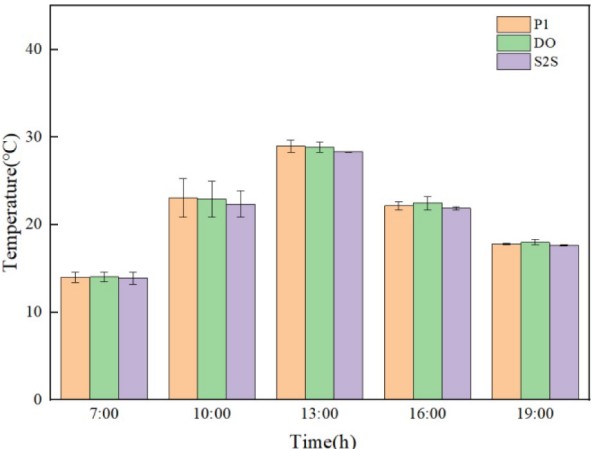

**Fig 11. Comparison error diagram of temperature simulation results and experimental results for the three radiation models (t7 position).**

temperature change during this time period. The numerically simulated result of temperatures by P1, DO, and S2S radiation models are 13.18°C, 12.41°C, and 12.91°C respectively around 7:30. The temperature error between the experimental and simulation data is 3.28°C, 2.51°C and 3.01°C. After the insulation is removed at 7:30 in the morning, as a result of the enhancement of outdoor temperature and solar radiation, the indoor air temperature is rapidly increased and reached 35.3°C at 12:30. The slope of the measured experimental data from the beginning of the temperature rise to the highest temperature is 4.62°C/h. The slopes of the P1, DO, and S2S radiation models from the beginning of heating up to the highest temperature are 4.1°C/h, 4.11°C/h and 3.93°C/h respectively. The P1 and DO models rise at about the same rate, reaching the maximum temperature at basically the same time. S2S model is followed. The highest temperature of simulated result in P1, DO and S2S radiation models are 35.73°C, 35.03°C and 34.55°C respectively. The temperature error between experimental and simulation data is 0.43°C, -0.27°C and -0.75°C respectively. From 13:00 to 16:00, the outdoor solar radiation is rapid decreased, the temperature of the indoor air is also showed a rapid

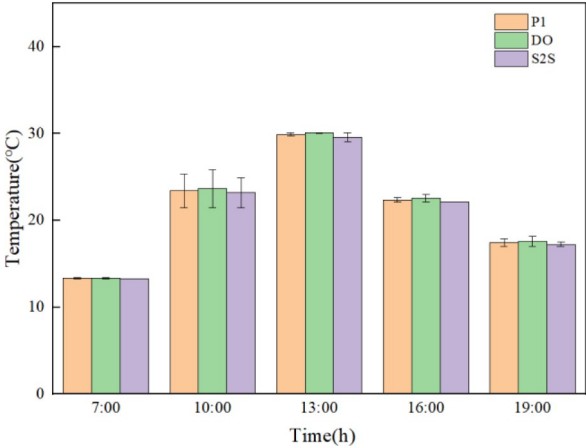

**Fig 12. Comparison error diagram of temperature simulation results and experimental results for the three radiation models (t5 position).**

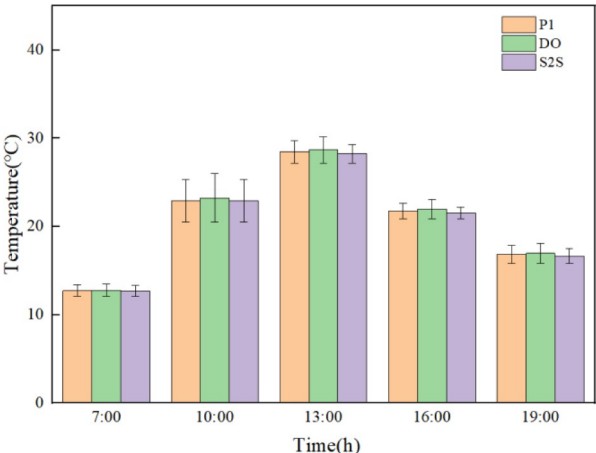

**Fig 13. Comparison error diagram of temperature simulation results and experimental results for the three radiation models (t2 position).**

downward trend. When the insulation blanket is recovered at 16:00, the downward trend is performed to slow down.

The error analysis of simulation and experimental results are shown in Fig 15. The maximum and minimum error analysis is shown in Table 5. The air temperature profile inside the greenhouse fitted by the DO radiation model best matches the trend of the experimentally measured temperature profile. The maximum error of the simulation results is 25.33%, and the minimum error of the simulation results is -0.76%.For the indoor air of the greenhouse, the temperature predicted by the DO model is more in line with the measured temperature than other models.

The temperature distributions of typical greenhouse profiles for the three radiation models at different moments are shown in Figs 16–18 respectively. During the day, the solar radiation is strong, the greenhouse plastic film temperature is the highest, from the plastic film to the indoor air, back wall and soil is decreasing state. Greenhouse enclosure is affected by cold air outside the greenhouse so the temperature of greenhouse enclosure and ground is lower. In the greenhouse at night, the heat dissipation of each enclosure becomes the main factor of heat loss in the greenhouse. Because the heat transfer coefficient of each surface is not very different, the indoor temperature is evenly distributed. The temperature

**Table 4. Error analysis under different radiation models at different locations.**

| position | models | Maximum absolute error | Minimum absolute error |
|----------|--------|------------------------|------------------------|
|          | P1     | 5.56°C                 | 0.21°C                 |
| T2       | DO     | 5.5°C                  | 0.59°C                 |
|          | S2S    | 3.93°C                 | 0.15°C                 |
|          | P1     | 4.41°C                 | 0.06°C                 |
| T5       | DO     | 5.05°C                 | 0.04°C                 |
|          | S2S    | 4.15°C                 | 0.03°C                 |
|          | P1     | 4.72°C                 | 0.02°C                 |
| T7       | DO     | 4.25°C                 | 0.52°C                 |
|          | S2S    | 3.54°C                 | 0.04°C                 |

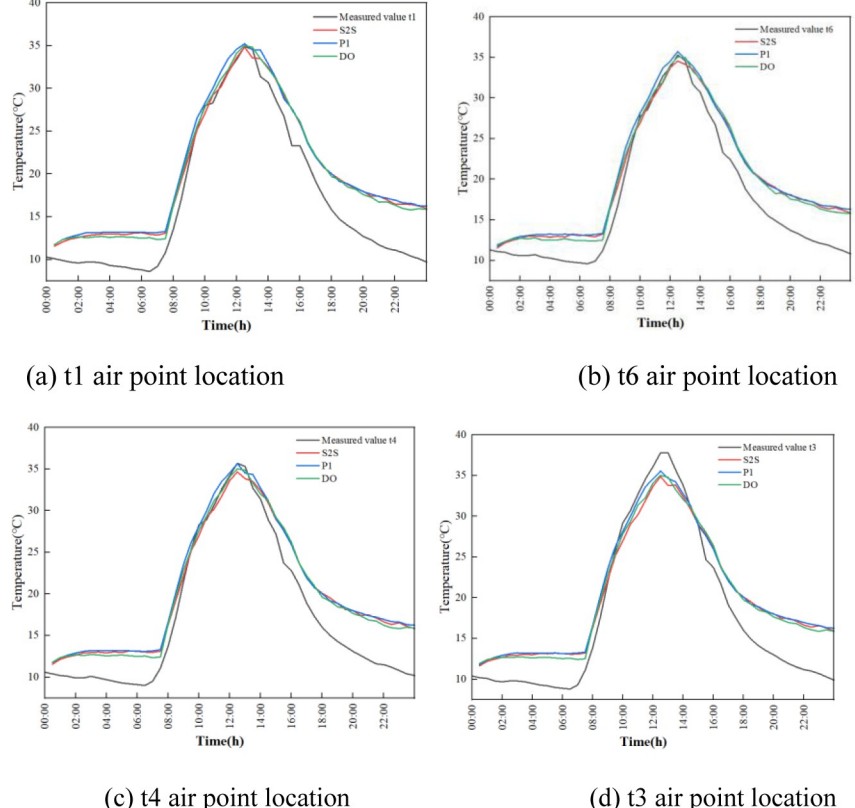

(a) t1 air point location

(b) t6 air point location

(c) t4 air point location

(d) t3 air point location

**Fig 14. Experimental and simulated temperature time plots for different positions and radiation models.**

of the place near the plastic film is relatively low, because the heat transfer coefficient of the insulation is larger than that of the back wall and the back slope. But the overall indoor temperature distribution is not very different.

## Discussion

In this study, the computational fluid dynamics method is used to explore the thermal radiation model applicable to the internal environment of greenhouses. The results of this study suggest that thermal radiation models should not be ignored if accurate results are to be obtained. Different thermal radiation models fit different temperature profiles.

Although a large number of studies have been reported on the greenhouse thermal radiation model [33, 34], these studies have not explore the specific thermal radiation models that could provide more accurate numerical simulations of greenhouse climate conditions. The energy storage wall temperature, the soil temperature and the indoor air temperature are often used as three verification objects of the thermal environment model [35, 36], but there are few studies on the simultaneous simulation and validation of the three. CFD can establish a complete simulation model of CSG based on mass, momentum, and energy conservation equations, as well as other model equations to calculate and obtain simulation results for the temperature field at any point in a greenhouse [37–39]. In this study, the thermal environment model of CSG was established by the computational fluid dynamics method. The input data of the model were the temperature of the air outside the greenhouse, the thermal power of the

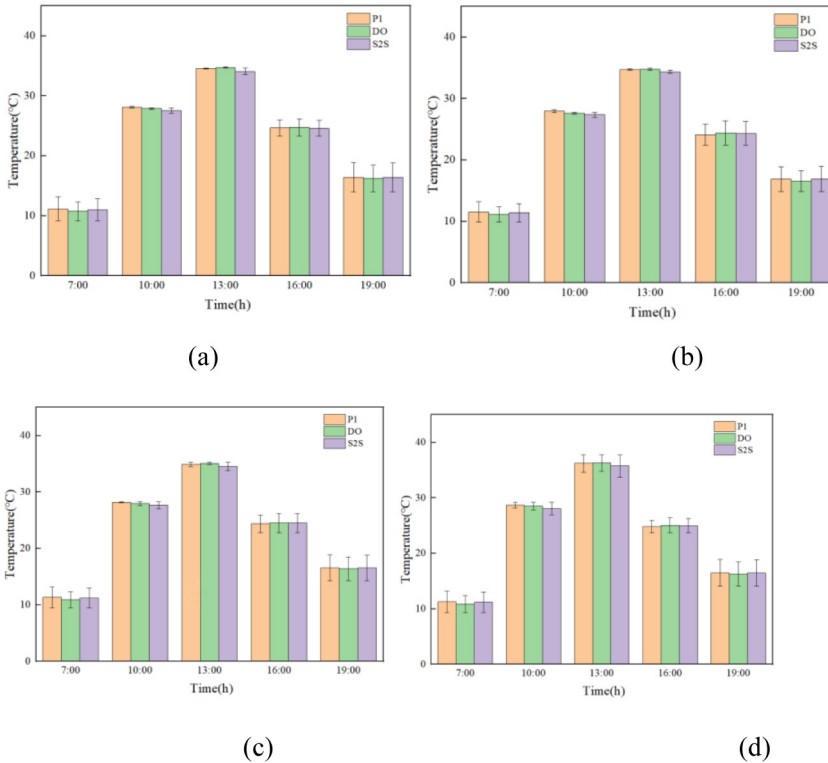

**Fig 15. Comparison error diagram of temperature simulation results and experimental results for the three radiation models.** (a) t1 point location; (b) t6 point location; (c) t4 point location; (d) t3 point location.

solar radiation and so on. The output data were the temperature of the indoor air, the temperature of the indoor rear wall and the temperature of the indoor soil.

Although thermal radiation models for predicting the greenhouse environment and the distribution of the greenhouse temperature were analyzed, applicability of radiation models under other weather conditions and humidity distribution in a greenhouse also could be

**Table 5. Error analysis under different radiation models at different locations.**

| position | models | Maximum absolute error | Minimum absolute error |
|---|---|---|---|
|  | P1 | 4.24˚C | 0.25˚C |
| T1 | DO | 3.9˚C | 0.12˚C |
|  | S2S | 4.81˚C | -0.06˚C |
|  | P1 | 3.68˚C | -0.04˚C |
| T4 | DO | 4.16˚C | -0.06˚C |
|  | S2S | 4.1˚C | -0.21˚C |
|  | P1 | 3.99˚C | 0.24˚C |
| T6 | DO | 2.51˚C | 0.27˚C |
|  | S2S | 4.08˚C | -0.05˚C |
|  | P1 | 4.32˚C | 0.2˚C |
| T3 | DO | 4.3˚C | 0.05˚C |
|  | S2S | 4.11˚C | 0.11˚C |

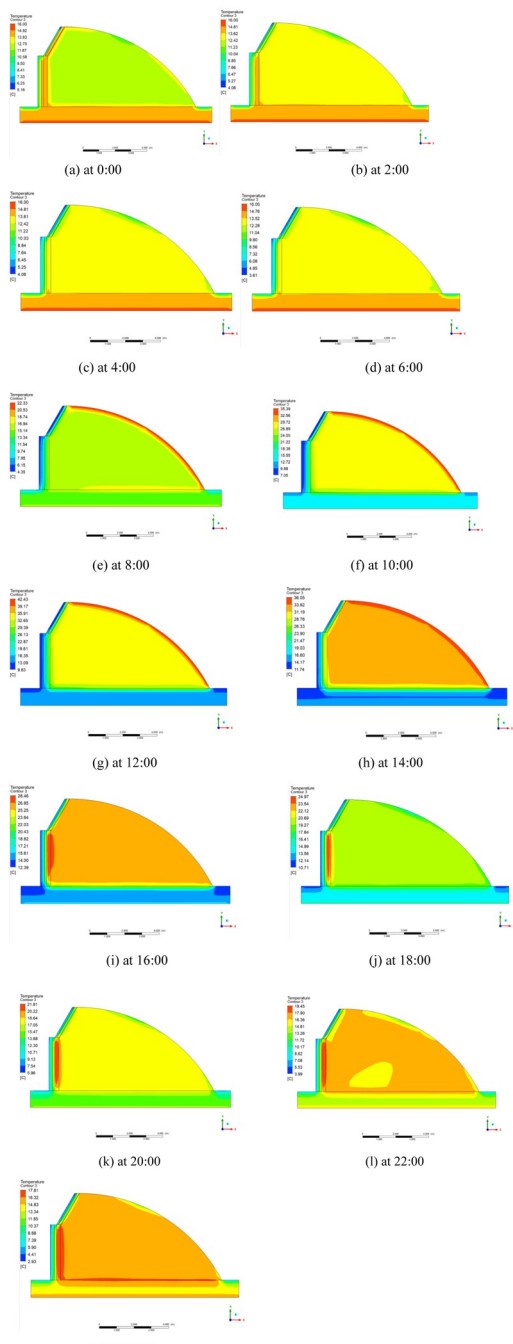

(a) at 0:00

(b) at 2:00

(c) at 4:00

(d) at 6:00

(e) at 8:00

(f) at 10:00

(g) at 12:00

(h) at 14:00

(i) at 16:00

(j) at 18:00

(k) at 20:00

(l) at 22:00

(m) at 24:00

**Fig 16. Temperature distribution of DO radiation model from 0:00–24:00.**

studied in the future. Due to the limited experimental conditions, the number of spatial distribution points is relatively small. These deficiencies will be further supplemented and improved in future research.

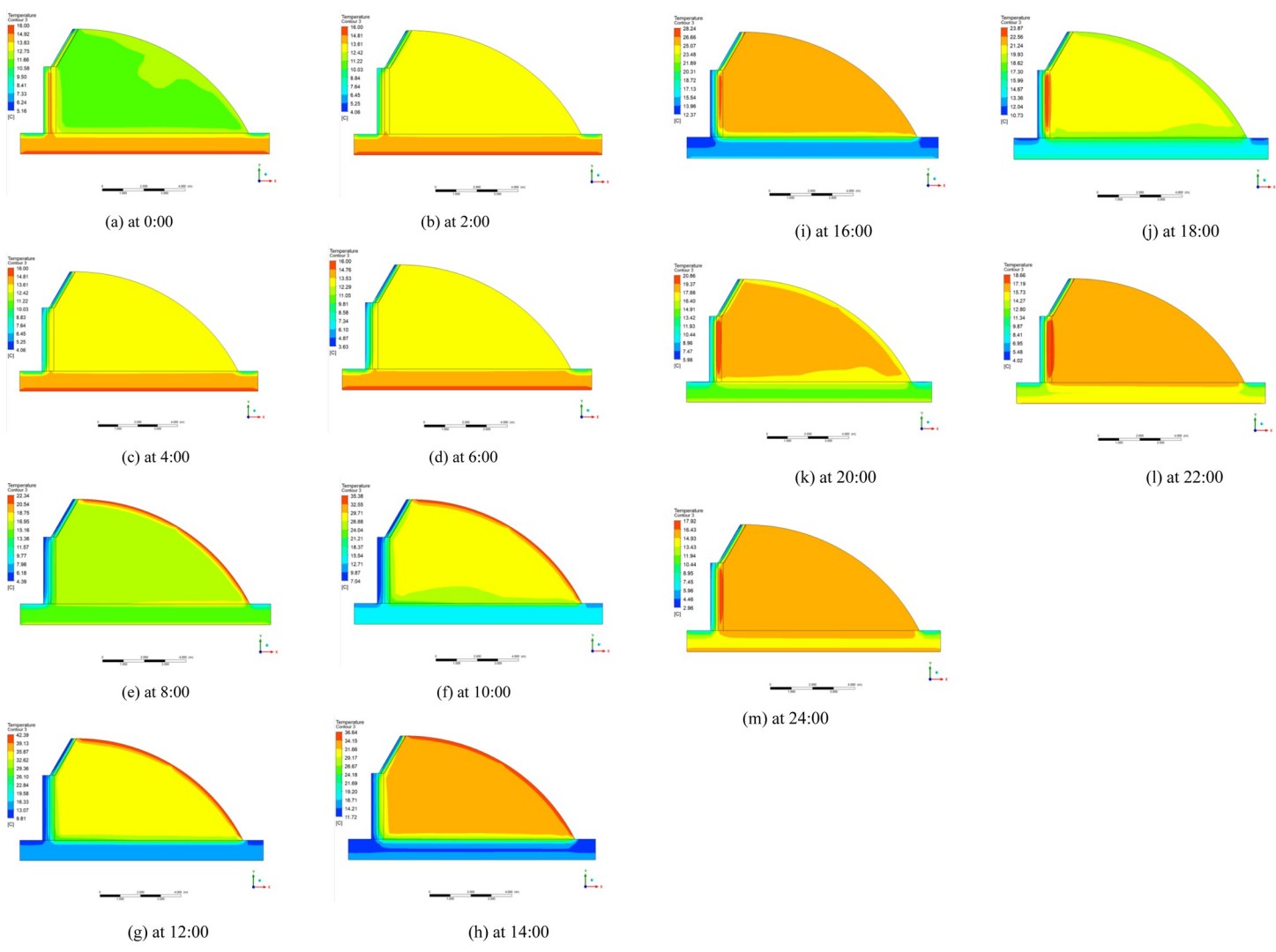

**Fig 17. Temperature distribution of S2S radiation model from 0:00–24:00.**

## Conclusion

The accuracy of P1, DO and S2S radiation models is investigated to predict the temperature of the rear wall, soil and air inside the heliostat. For the indoor rear wall of the greenhouse, the P1 model shows the best accuracy with a maximum and minimum temperature difference of 3.24˚C and 0.22˚C between the experimental data and the simulation results. For soil in the greenhouse, the S2S model shows the best accuracy with an average maximum and minimum temperature difference of 3.87˚C and 0.07˚C between experimental data and simulation results. For indoor air temperature in the greenhouse, the DO model shows the best accuracy with an average maximum and minimum temperature difference of 3.72˚C and 0.125˚C

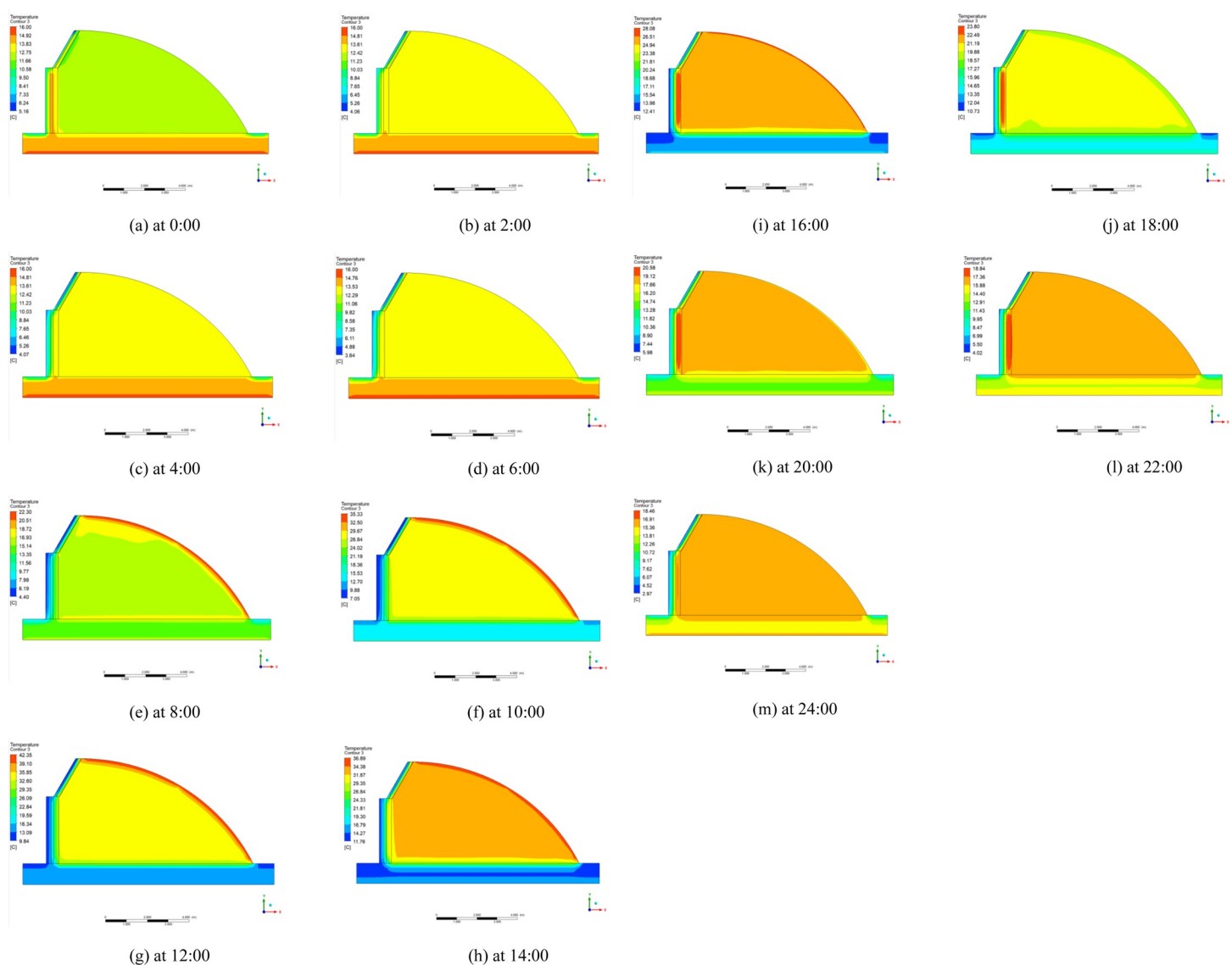

**Fig 18. Temperature distribution of P1 radiation model from 0:00–24:00.**

between experimental data and simulation results. It is thus clear that the selection of a suitable thermal radiation model plays an important role in predicting the temperature aspects. The results of the study provide useful information for the selection of thermal radiation in greenhouses and have important implications for temperature monitoring and control in precision agriculture.

## Supporting information

**S1 File.**
(XLSX)

## Author Contributions

**Data curation:** Zehui Li.

**Methodology:** Simin Cao.

**Resources:** Zhanyang Xu.

**Writing – original draft:** Wenhe Liu, Manhe Qu.

**Writing – review & editing:** Feng Zhang.

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
