## [Decision Letter · Decision Letter 0]

17 Jan 2024

PONE-D-23-28972A Comparative Study of Thermal Radiation Model in Chinese Solar GreenhousePLOS ONE

Dear Dr. Zhang,

Thank you for submitting your manuscript to PLOS ONE. After careful consideration, we feel that it has merit but does not fully meet PLOS ONE’s publication criteria as it currently stands. Therefore, we invite you to submit a revised version of the manuscript that addresses the points raised during the review process as given below:

I am returning your manuscript with two reviews. After reading the reviews and looking at the manuscript, I am afraid that I have to concur with the critical reviews as provided by the reviewers.

Specifically, I have a few major observations on the overall presentation of the work in the manuscript including the clarity in demonstrating the need of the study, critical review from the literature, parametric specifications, the method of evaluation with adequate justification, Grammar and spelling corrections.

I am sorry I cannot be more positive at the moment. However, as I have noted, all is not lost. It requires a lot of work and a major revision that I believe you need more time to work on the manuscript for a resubmission if you wish to do so. Please note that it will have to go through the second round of review. Hence, pay attention to the abovementioned suggestions, the following reviewer suggestions and give them due consideration when revising your manuscript.

You must provide all data underlying the findings in your manuscript fully available for the review process without any restrictions. The data should be provided as part of the manuscript or its supporting information, or deposited to a public repository. If there are restrictions on publicly sharing data—e.g. participant privacy or use of data from a third party—those must be specified. Otherwise your paper cannot be accepted.

We encourage you to submit your revised manuscript by Mar 02 2024 11:59PM. If you will need more time than this to complete your revisions, please reply to this message or contact the journal office at plosone@plos.org. Please include the following items when submitting your revised manuscript:A rebuttal letter that responds to each point raised by the academic editor and reviewer(s). You should upload this letter as a separate file labeled 'Response to Reviewers'.A marked-up copy of your manuscript that highlights changes made to the original version. You should upload this as a separate file labeled 'Revised Manuscript with Track Changes'.An unmarked version of your revised paper without tracked changes. You should upload this as a separate file labeled 'Manuscript'.

We look forward to receiving your revised manuscript.

Kind regards,

Rajagopalan Parameshwaran, Ph.D.

Academic Editor

PLOS ONE

4. In your Methods section, please provide additional information regarding the permits you obtained for the work. Please ensure you have included the full name of the authority that approved the field site access and, if no permits were required, a brief statement explaining why.

5. Thank you for stating the following in the Funding Section of your manuscript:

“Key Research and Development Projects of Liaoning Province

(No.2021JH210200022), China Postdoctoral Science Foundation Funded Project

(No.2021M693862).”

6. We note that your Data Availability Statement is currently as follows: [All relevant data are within the manuscript and its Supporting Information files.]

7. PLOS requires an ORCID iD for the corresponding author in Editorial Manager on papers submitted after December 6th, 2016. Please ensure that you have an ORCID iD and that it is validated in Editorial Manager. To do this, go to ‘Update my Information’ (in the upper left-hand corner of the main menu), and click on the Fetch/Validate link next to the ORCID field. This will take you to the ORCID site and allow you to create a new iD or authenticate a pre-existing iD in Editorial Manager. Please see the following video for instructions on linking an ORCID iD to your Editorial Manager account: https://www.youtube.com/watch?v=_xcclfuvtx

Reviewers' comments:

Reviewer's Responses to Questions

**Comments to the Author**

1. Is the manuscript technically sound, and do the data support the conclusions?

Reviewer #1: Partly

Reviewer #2: No

2. Has the statistical analysis been performed appropriately and rigorously? 

Reviewer #1: Yes

Reviewer #2: No

3. Have the authors made all data underlying the findings in their manuscript fully available?

Reviewer #1: Yes

Reviewer #2: No

4. Is the manuscript presented in an intelligible fashion and written in standard English?

Reviewer #1: No

Reviewer #2: Yes

5. Review Comments to the Author

Reviewer #1: Dear Sir/Madam

The manuscript should be revised based on the comments that reviewers provided to ensure that minimum requirements of the research paper is satisfied. Please check the attachment for details.

Regards.

Reviewer #2: 1.An overarching problem and its solution need to be explored in order to validate the consideration of Chinese Solar Greenhouse (CSG) for research.

2.The introductory section should delve into the scientific principles underlying the functioning of the Chinese Solar Greenhouse (CSG).

3.The author's objective in employing three models to identify the optimal conditions for crop growth is communicated. However, there is a need for elaboration on the novelty statement, specifically for what particular crop thermal environment is validated? what is the appropriate environment for the crop considered? the reference source stating the best environment for the crop considered etc. Only with these considerations can the correctness of experimental and simulated validations be established.

4.In table 1 not all the building materials are mentioned. Is the CSG built without any steel, iron or aluminum materials.

5.Green house dryer works best at forced convection. Is the RNG model used for turbulence considering forced convection?

6.Attach experimental set up images, images of crops dried, specifications of temperature sensors used, heights in which the sensors are positioned, calculate their uncertainty, why the experimental values and experimental part is not been discussed in the manuscript?

7.From my understanding the manuscript validates the data collected in experiment using three models in fluent. The authors claim that your validating the best thermal environment for crop growth, without telling what crop you used in experiment and a source data mentioning the appropriate environment for that particular crop, how the validation can be said correct?

6. PLOS authors have the option to publish the peer review history of their article (what does this mean?). If published, this will include your full peer review and any attached files.

Reviewer #1: No

Reviewer #2: Yes

---

## [Author Response · Author response to Decision Letter 0]

4 May 2024

We sincerely thank the editor and all reviewers for their valuable feedback that we have used to improve the quality of our manuscript. According to your nice suggestions, we have made extensive corrections to our previous manuscript, the detailed corrections are listed below.

Reviewer #1: 

1. Which radiation model was used in Ref [10] CFD model?

The radiation model used for CFD in reference [10] was added. The modified content is as follows: Zhang Fang et al. [10] established a large-span greenhouse based on CFD technology and used DO radiation model to simulate the temperature and airflow field under natural ventilation conditions.

2. What is the meaning of “radiative effective” mentioned from Ref [12] results?

We were really sorry for our careless mistakes. “radiative effective” should be “radiative effects”. It is demonstrated that the inclusion of thermal radiation in the CFD model is vital as the air temperature in the lower levels can be underpredicted if the radiative effects are ignored. The modified content is as follows: The radiative effects were introduced in the thermal flowing models as a critical condition for indoor temperature environments.

3. The turbulence models should be typed like formulas and “by compared with” needs revision in the Ref [14] results.

We sincerely appreciate the valuable comments. We have checked the literature carefully and recomposed Ref [14], which is now Ref [8]. The turbulence models typed like formulas and “by compared with” were added in 2.4. The modified content is as follows:

The performance of three widely used turbulence models, the standard k–e model (SKE), the renormalization group k–e model (RNG), the realizable k–e model (RKE), were compared for their ability to predict the airflow velocities and ammonia concentrations in the scale model swine building enclosure. The RNG model was suitable for predicting weak airflows. The velocities and concentrations predicted by the RNG model were closer to the measured values than other two models [8].

The transport equations and model constants for the turbulence models are listed in Table 3[8].

Table 3.Transport equations and model constants for the turbulence models [8].

Model Transport equations Model constants

SKE =1.44

 =1.92

RNG =1.42

 =1.68

RKE =1.44

 =1.9

4. The sentence “Even if the using of CFD has been widely expanded to analysis this problem in depth.” should be modified.

We have modified this sentence based on the contexts. The modified content is as follows:

During the research of designing parameters’ effect on the thermal environment in indoor CSG, the use of CFD codes has been extended to gain insight into this problem, but selecting appropriate sub-models for the influence of convection, radiation and turbulence were still a huge challenge [17].

5. Generally, section 2, first paragraph “Experimental method and theoretical considerations” needs a revision.

We think this is an excellent suggestion. We have added and modified the first paragraph of section 2. The modified content is as follows:

To validate the CFD model, real-scale experimental data were used. The solar greenhouse used is located in Northeast China (latitude:41°49′N, longtitude:123°34′). CSG is composed of north wall, east wall, west wall, plastic film and ground soil without heating measurement in winter. The greenhouse is a length of 60m, a span of 8m, a rear wall height of 3.2m and a ridge height of 5m respectively. The rear wall is composed of 240mm clay bricks, 120mm polyethylene benzene board, and 240mm clay bricks. The rear slope is composed of 15mm wooden boards, 150mm polyethylene benzene board, 25mm cement, and 90mm waterproof layer. The front roof is made of 0.1mm PVC anti-aging plastic film for daytime lighting and heat storage. At night, the insulation covering materials are 1.5mm PE woven fabric, 27mm spray glue cotton, and 1.5mm PE woven fabric from the inside out. The spray glue cotton plays a role in insulation, while the inner and outer layers of PE woven fabric play a waterproof role. It is opened at 8:00 and covered at 16:00 in every day. Fig 1 shows the indoor and outdoor conditions of the experimental greenhouse. 

Fig 1. Experimental solar greenhouse

6. The mesh convergence study should be added.

We sincerely appreciate the valuable comments. In 2.3, the convergence of the mesh was supplemented. The modified content is as follows:

Under the same numerical simulation conditions, simulation calculations were conducted and point T4 was selected as the temperature verification feature point. The consistency and stability of the simulation results were compared. The relationship between the simulated temperature of the feature point and the grid size is shown in Figure 5. When the grid size is less than 0.3m, the temperature of the feature point fluctuates significantly. When the grid size is greater than 0.3m, the temperature of the feature point does not change significantly with the increase of the grid size and remains relatively stable.

7. The sentence “The outdoor temperature is timely by TRM-ZSF GPRS wireless remote control system of Jinzhou Sunshine Technology Co., Ltd. Monitors”, requires correction.

We have revised this sentence to “The TRM-ZSFGPRS wireless remote control system from Jinzhou Sunshine Technology Co., Ltd. is used to monitor outdoor temperature”.

8. According to Fig. 4, what is the logic of temperature probes arrangement and why 8 probes were selected? What is the reason for putting the ambient temperature probe at that specific location? Did it sense the outside wind effect?

We have explained and supplemented this section according to the reviewer's suggestions. The modified content is as follows: The temperature sensors are arranged using the cross-sectional method. In the greenhouse, four temperature measurement points are arranged in two layers on a section 5m away from the south wall. Three temperature measurement points (T1, T4, T6) are arranged at a height of 1m from the ground on the bottom layer, and one temperature measurement point (T3) is arranged at a height of 2m from the ground on the upper layer. Three temperature measurement points (T2, T5, T7) are arranged on the soil inside the greenhouse. One rear wall temperature measurement point (T8) is arranged at a height of 1.6m from the ground, and the temperature measurement points are basically not affected by external wind.

9. The governing equations should be mentioned and the relation between source term of energy equation with radiation models should be described.

In 2.3, governing equations and relation between source term of energy equation with radiation models were added. The modified content is as follows:

Simulating the distribution of temperature and airflow field, etc. in a greenhouse, the compressibility of a gas moving at low speeds can usually be solved as an incompressible fluid when it does not have much effect on its motion and equilibrium problems. The air flow rate in the solar greenhouse is a low-flow rate flow, and the size of the flow rate is usually around 0.1~0.5m/s. Therefore, the basic control equations satisfied by the fluids in the greenhouse include the mass conservation equation, the momentum conservation equation, the energy conservation equation, and the generalized form of the control equations.

(1) Continuity equation:

 (1)

where ρ is the density of the gas. represents the component of the velocity vector in the x,y,z direction. is the mass of the secondary phase added to the continuum term, which can be any custom source term.

(2) Conservation of momentum equation:

 (2)

 (3)

 (4)

where is the pressure on the fluid micromeres in Pa. is the viscous stress on the micromeres. is the different micromeres viscous components in the corresponding.

(3) The conservation of energy equation:

 (5)

where is the effective heat transfer coefficient, . is the turbulent heat transfer coefficient, is the total energy. is the change in enthalpy of the humid air transport process. is the volumetric heat source term.

10. The sentence “The large scale turbulence and the widest application range in industrial flow and heat transfer simulation are reasonably”, needs revision.

We have revised this sentence to “It can reasonably predict large-scale turbulence and has the widest application range in industrial flow and heat transfer simulation”.

11. The authors need to explain how the ideal gas can be considered as incompressible and with this assumption, the buoyancy effect can be simulated.

Simulating the distribution of temperature and airflow field, etc. in a greenhouse, the compressibility of a gas moving at low speeds can usually be solved as an incompressible fluid when it does not have much effect on its motion and equilibrium problems. The air flow rate in the solar greenhouse is a low-flow rate flow, and the size of the flow rate is usually around 0.1~0.5m/s.

12. What does this sentence mean? “The DTRM radiation model is applicable to all optical thickness models, but it is rarely used because it cannot be solved”.

We have modified this sentence into “The DTRM model works for all optical thicknesses, but does not take into account the scattering effect of the radiation in the calculation, which increases the burden on the CPU if ray tracing is to be used.”

13. The explanations about S2S radiation model should be more carefully rewritten.

Thank you for the suggestion. We have added the information required as explained about S2S radiation model. The modified content is as follows: The full name of the S2S radiation model is Surface to Surface radiation model, which is a model that only considers thermal radiation between surfaces. It is a radiation model suitable for zero optical thickness scenarios. If the medium in the computational domain is some medium with strong absorption of thermal radiation, such as water vapor (polyatomic molecules), it is not suitable to use this model. This model is more suitable for air or diatomic molecules, etc.

14. The number of discretization in each direction for DO model should be mentioned and their independency should be studied.

We have added discretized setup parameters to the DO model. The modified content is as follows: The more discretization times, the more accurate the simulation results will be, but the computational complexity will also increase. This study used a 3D model to solve a total of 8 quadrants with values theta divisions and phi divisions set to 2. The research on its independence is beyond the scope of this article, so it cannot be modified at present. However, we are willing to explore and further study your suggestions in future work.

15. The optical boundary conditions for modeling radiation, convention and conduction are required which are the main parts that should be added.

In 2.4, the optical boundary conditions for modeling radiation, convention and conduction are added. The modified content is as follows:

Import the grid file into the computational fluid dynamics software ANSYS Fluent 2022R2 for numerical solution. Fluent provides rich computational models and uses the finite volume method to solve the computational domain. In this study, the greenhouse film is treated as semi-transparent boundaries, and the soil is treated as opaque boundaries. The absorption and transmittance of solar radiation on each surface are set, and specific optical parameters are shown in Table 2. The radiation boundary conditions for greenhouse plastic films need to consider the impact of environmental radiation. Based on experimental data, the benchmark temperature for the early morning environment is set at 11°C, and the solar radiation factor for March was determined based on historical meteorological data and relevant research in Shenyang, to match the simulated and measured temperature time series. In addition, boundary conditions for greenhouse simulation are set in combination with actual measurement conditions. The external temperature values of the greenhouse at different times are used as the environmental convection boundaries for the enclosure structure and outdoor soil. The temperature values are shown in Fig. 6. Using temperature values at different times as the environmental convection boundary of the greenhouse film, the temperature values are shown in Fig. 7. In order to solve the heat transfer process of solar greenhouse under 24h sunlight, the solution step is set as 1800s, and the total solution time is 86400s.

Table 2.Optical parameters of materials in greenhouses

materials/position Absorption coefficient(m-1) emissivity absorptivity transmissivity

air 0.50 - - -

Greenhouse film - 0.85 0.95 0.95

Soil surface - 0.95 0.6 -

Fig 6. Ambient temperature outside the greenhouse

Fig 7. Greenhouse plastic film temperature

16. What is the reason of differences between radiation models results and the experiments?

The possible reasons for errors are as follows: 1. Simulation models are usually established based on specific assumptions, which may differ from the actual situation. For example, the model may overlook certain details in the actual system, resulting in simulation results that are not completely consistent with the actual results. 2. The parameters in the simulation model are obtained through observation or experimentation, and these data may have random errors.

17. “is lowed” should be corrected with “is reduced”.

We sincerely thank the reviewer for careful reading. As suggested by the reviewer, we have corrected the “is lowed” into “is reduced”. The modified content is as follows: The overall measured temperature at t2 is reduced due to the distance of the adjacent greenhouse, which is shaded by solar radiation.

18. According to Fig. 11, the peak of t3 is higher than t4 which is not observed in the temperature contours of figures 13,14 and 15. Please describe the reason.

According to Fig. 11, the peak of t3 is higher than t4 is measured value. Their simulated values do not differ significantly, while Figures 13, 14, and 15 show the simulated temperature cloud maps.

19. The manuscript should be spell checked and grammar checked again.

Thanks for your suggestion. We have tried our best to polish the language and check the spell in the revised manuscript.

Reviewer #2: 

1. An overarching problem and its solution need to be explored in order to validate the consideration of Chinese Solar Greenhouse (CSG) for research.

Due to cost limitations of greenhouses, most greenhouses in Northeast China are single slope sunlight greenhouses, and the temperature inside the sunlight greenhouses is an important indicator affecting crop growth. Due to the influence of solar radiation, the temperature inside the greenhouse is too high at noon, often exceeding the suitable growth temperature for crops. This study conducted CFD numerical simulations on solar greenhouses in Northeast China and analyzed the effects of three radiation models (P1, S2S, and DO) on predicting the temperature environment inside the greenhouse. The purpose is to study the ability of different radiation models to predict the temperature environment inside greenhouses, in order to make more accurate numerical predictions of the climate conditions inside solar greenhouses.

2. The introductory section should delve into the scientific principles underlying the functioning of the Chinese Solar Greenhouse (CSG).

The operation principle of a solar greenhouse is mainly based on the utilization of solar energy. It allows sunlight to enter the interior of the greenhouse through transparent materials such as glass or plastic. These materials not only allow light to pass through, but also absorb incoming light energy and convert it into thermal energy, thereby increasing the temperature inside the greenhouse. During the day, sunlight shines into the greenhouse, and transparent materials accumulate heat. At night, when the outdoor temperature drops, the insulation of the greenhouse will be turned off to reduce the loss of indoor heat. Meanwhile, relying on the slow release of heat from walls and soil at nig

---

## [Decision Letter · Decision Letter 1]

25 Jul 2024

PONE-D-23-28972R1A Comparative Study of Thermal Radiation Model in Chinese Solar GreenhousePLOS ONE

Dear Dr. Zhang,

Thank you for submitting your manuscript to PLOS ONE. After careful consideration, we feel that it has merit but does not fully meet PLOS ONE’s publication criteria as it currently stands. Therefore, we invite you to submit a revised version of the manuscript that addresses the points raised during the review process. Please find the reviewer comments as an attachment herewith.

We look forward to receiving your revised manuscript.

Kind regards,

Rajagopalan Parameshwaran, Ph.D.

Academic Editor

PLOS ONE

Journal Requirements:

Reviewers' comments:

Reviewer's Responses to Questions

**Comments to the Author**

1. If the authors have adequately addressed your comments raised in a previous round of review and you feel that this manuscript is now acceptable for publication, you may indicate that here to bypass the “Comments to the Author” section, enter your conflict of interest statement in the “Confidential to Editor” section, and submit your "Accept" recommendation.

Reviewer #1: (No Response)

2. Is the manuscript technically sound, and do the data support the conclusions?

Reviewer #1: (No Response)

3. Has the statistical analysis been performed appropriately and rigorously? 

Reviewer #1: (No Response)

4. Have the authors made all data underlying the findings in their manuscript fully available?

Reviewer #1: (No Response)

5. Is the manuscript presented in an intelligible fashion and written in standard English?

Reviewer #1: (No Response)

6. Review Comments to the Author

Reviewer #1: (No Response)

7. PLOS authors have the option to publish the peer review history of their article (what does this mean?). If published, this will include your full peer review and any attached files.

Reviewer #1: No

---

## [Author Response · Author response to Decision Letter 1]

31 Jul 2024

Response to reviewer #1

We sincerely thank the editor and all reviewers for their valuable feedback that we have used to improve the quality of our manuscript. According to your nice suggestions, we have made extensive corrections to our previous manuscript, the detailed corrections are listed below.

Most of the grammar and spelling errors have been corrected; however, the authors have not addressed the following points. 

In question 5, the experimental set up were explained and it was mentioned that different materials were used in the solar greenhouse walls which their thermal and radiative boundary conditions were not given in Table 2. Although some assumptions can be made and average boundary conditions can be used; however, none was given in Table 2.

The Table 2 was modified as reviewer’s advice, which is shown as follow:

Table 2.Optical parameters of materials in greenhouses

materials/position Boundary type Absorption coefficient(m-1) emissivity absorptivity transmissivity

air transparent 0.50 - - -

Greenhouse film semi-transparent - 0.85 0.95 0.95

Soil surface opaque - 0.95 0.6 -

their thermal and radiative boundary conditions for greenhouse simulation are set in combination with actual measurement conditions. The external temperature values of the greenhouse at different times are used as the environmental convection boundaries for the enclosure structure and outdoor soil. The temperature values are shown in Fig. 6. Using temperature values at different times as the environmental convection boundary of the greenhouse film, the temperature values are shown in Fig. 7. In order to solve the heat transfer process of solar greenhouse under 24h sunlight, the solution step is set as 1800s, and the total solution time is 86400s.

In question 14 regarding the discretization in each direction for DO model, it was mentioned that the value for theta division was 8 and phi was 2 which are not enough at all. Then, the results of DO model are not reliable based on the author’s response.

According to the reviewer’s require, the value of DO model for theta division was modified as 8 and phi was modified as 5. Therefore, the results of DO model are reliable, which is modified in manuscript.

---

## [Editor Report · Decision Letter 2]

8 Aug 2024

A Comparative Study of Thermal Radiation Model in Chinese Solar Greenhouse

PONE-D-23-28972R2

Dear Dr. Zhang,

We’re pleased to inform you that your manuscript has been judged scientifically suitable for publication and will be formally accepted for publication once it meets all outstanding technical requirements.

Kind regards,

Rajagopalan Parameshwaran, Ph.D.

Academic Editor

PLOS ONE
---

## [Editor Report · Acceptance letter]

2 Sep 2024

PONE-D-23-28972R2 

PLOS ONE

Dear Dr. Zhang, 

I'm pleased to inform you that your manuscript has been deemed suitable for publication in PLOS ONE. Congratulations! Your manuscript is now being handed over to our production team.

Kind regards, 

on behalf of

Prof. Rajagopalan Parameshwaran 

Academic Editor

PLOS ONE